# Finnish Pre-Service Teachers’ Perceptions of Perceived Competence in Early Childhood Physical Education

**DOI:** 10.3390/ijerph18126454

**Published:** 2021-06-15

**Authors:** Anne Soini, Anthony Watt, Arja Sääkslahti

**Affiliations:** 1Department of Education, University of Jyväskylä, 40014 Jyväskylä, Finland; 2College of Arts and Education, Victoria University, Melbourne, VIC 8001, Australia; anthony.watt@vu.edu.au; 3Faculty of Sport and Health Sciences, University of Jyväskylä, 40014 Jyväskylä, Finland; arja.saakslahti@jyu.fi

**Keywords:** early childhood education teacher training, physical education, pre-service teacher, perceived competence

## Abstract

Early childhood education and care (ECEC) teachers have a central role in supporting young children’s physical activity (PA) and overall development in the early years. However, the value of early childhood education teacher training (ECETT) programmes is not widely understood. This study aimed to investigate pre-service teachers’ perceptions of perceived competence when (1) supporting a child’s PA, (2) teaching PE, and (3) observing and assessing a child’s motor skills and PA. These self-evaluations were compared with a range of individual, educational, and behavioural characteristics. Final-year Bachelor degree pre-service teachers (*n* = 274; 54%) from seven universities in Finland participated in the self-report questionnaire. The results of the linear regression models showed that the relevant PE studies and previous experiences of pre-service teachers predicted higher perceived competence of supporting a child’s PA, teaching PE, and observing and assessing a child’s motor skills and PA. Thus, the study findings demonstrated how teacher training could positively influence perceptions and attitudes to increase a person’s perceived competence when implementing PE in the early years. Overall, results reinforce the importance of PE in ECETT, and the time devoted to this syllabus area should be maintained or increased.

## 1. Introduction

Lifelong physical activity (PA) patterns have been regularly reported as developing in early childhood [1,2,3,4]. Recent recommendations from the World Health Organisation [4] outline that young children should engage in a minimum of 180 min of PA and at most 60 min of screen-based sedentary time per day. Additionally, three hours of PA per day of any intensity is recommended in Finland [5]. However, it remains of global concern that children do not reach the levels of daily PA proposed in the international guidelines and within the Finnish preschool-age population e.g., [6,7]. Through the support of families, early childhood education and care (ECEC) settings are institutions that serve a critical role in increasing PA levels and improving the health and development of young children e.g., [8,9,10,11,12,13]. Within teacher education fields, it is important to evaluate the skills and knowledge of individuals as they progress through their professional preparation [14]. Examining pre-service teachers’ perceptions of their early childhood education teacher training (ECETT) will support programme development and, subsequently, quality of physical education (PE) delivery of in-service ECEC.

According to Howell and Sääkslahti [15], no standard specifications in early years PE curriculums have been described. Belgium and the UK encourage two hours a week of PE within the curriculums. In addition, Ireland and the UK have specific elements of the curricula dedicated to physical development or PE as primary areas of children’s learning. In Finland, China, Denmark, and Italy, the early years’ curriculum states the importance of children’s PA; however, it does not describe the amount of PA to be achieved within childcare [15]. Additionally, a recent study by Tortella et al. [16] has highlighted that Nordic countries have a strong outdoor culture. For instance, Finnish children in ECEC are engaging in approximately three hours per day of outdoor play. Moreover, outdoor activities are often considered to substantially contribute to a child’s daily PA [17], and outdoor play positively affects a child’s development and health [16].

In Finland, an essential goal of ECEC is to encourage children to engage in physical activities outdoors during all seasons. The National Core Curriculum for Early Childhood Education and Care [18] also emphasises the importance of adequate PA, both structured and free play, for a child’s healthy growth, development, learning, and well-being. In this respect, ECEC teachers act as PA promoters in childcare centres by organising systematic, goal- and child-focussed activities that support children’s PA and fundamental motor skills (FMS) development. The purpose of PE is to support the child’s physical, motor, cognitive, psychological, and socio-emotional growth at all age stages in early childhood. The activities can be guided, but the role also includes enabling children’s spontaneous PA. For example, adequate PA can be made possible for children by adapting the environment to better suit physical activities [5].

Teachers in ECEC have a vital role in supporting young children’s overall development and health in the early years. Based on the analysis and comparison of the ten countries’ national recommendations, the results indicate marked differences in staff education, knowledge, and skills on how to support children through physical activities that promote physical and motor development in early ages [15]. Supportive PA engagement behaviours by ECEC teachers are important because children spend a significant amount of their time with them [19]. Earlier studies have shown that the children attending ECEC centres with more resources and a broad range of levels of tertiary-educated teachers and workers demonstrated significantly higher levels of physical activity [20,21,22]. According to recent research by Mavilidi et al. [21], teachers’ training and professional development may provoke valuable changes in children’s daily routines to increase PA participation.

Previous PE studies in school PE teacher education (PETE) have indicated the positive relationship between professional development programmes and teachers’ knowledge, perceptions, and teaching behaviour, as well as the positive influence that teachers have on students’ learning performance [23]. The enhancement of ECEC teachers’ knowledge of children’s motor development can influence higher levels of PA in children [24,25]. Veldman et al. [26] indicated that in centres with higher intentional teaching practices, children spent significantly more time in moderate-to-vigorous-intensity physical activity (MVPA) compared to centres with lower intentionality. A child can have a very different gross motor experience even within the same setting and facility based on their teacher’s beliefs, creativity, and engagement level [27]. Lu and Montague’s [28] research highlighted that teachers’ involvement and enthusiasm significantly affect children’s participation in physical activities; see also [29]. Children’s PA levels are also higher when PE classes are taught by more physically active teachers [30]. It is also important to acknowledge that if a person enjoys an activity, finds it personally attractive, and develops the skills where he/she feels competent, there is an increased likelihood that he/she will continue to engage in those tasks [31]. Past experiences determine how a person perceives his or her current skills and competence. Finally, teachers’ participation in play with children may differ by gender; for instance, male teachers seem to have more play willingness and participate more in physically active play [32]. In contrast, female teachers tend to prioritise calm play.

In Finland, ECEC teachers with a bachelor level degree are responsible for the child’s pedagogical education. The Finnish education system is based on trust in teachers and teacher education; consequently, teachers have the pedagogical freedom within the curriculums to organise their teaching [33]. Furthermore, teachers can support a child by providing individual activities because the adult–child teaching ratio in Finland is 1:7. Although ECEC teachers’ initiations are positively related to children’s PA, e.g., [34], Finnish ECEC teachers rarely organise physical play or encourage children to engage in physically active play [35]. According to the report of Repo et al. [36], PE in Finland is mainly well implemented in ECEC; however, nearly one-fifth of respondents indicated that daily vigorous PA is not available. Additionally, a study [37] has shown that children’s individual expression through their bodies (e.g., drama plays and dance) is more consistently realised in the children’s free plays than teacher-led activities. It is noted that staff perceive their pedagogical skills as lacking in the domain of children’s bodily expression [37].

A variety of job roles exist within the ECEC sector, and the requirements of childcare staff differ significantly from country to country [15,38]. Nevertheless, some broad groups of child care providers can be identified. Many teachers have been trained in Nordic and central European countries in upper-secondary or tertiary education, focussing on early childhood services rather than primary teaching [38]. In Finland, since 1995, the degree of ECEC teacher has been a university-level bachelor’s degree in education. Currently, seven Finnish universities (from south to north: University of Helsinki, University of Turku, University of Tampere, University of Jyväskylä, University of Eastern Finland, Åbo Akademi, and University of Oulu) offer three years of studies (180 credits, ECTS) leading to the Bachelor of Education. A bachelor’s level education qualifies one to work in both ECEC and pre-primary education for 6-year-olds. In addition, all of these universities also offer courses leading to the Master of Education (120 cr, two years). A masters’ degree is required for the directors of ECEC settings.

All seven universities providing ECETT in Finland can create their syllabuses, leading to minor variation in the content and specialities. The bachelor’s degree syllabus structure includes General studies, Communication and language studies, Basic Studies, Intermediate Studies, Professional Studies (e.g., PE studies), and Elective Studies. There is no specific PE curriculum specification, and therefore, universities can independently decide on the amount and content of PE studies offered. However, it is notable that Finland is including outdoor activities in its ECETT programmes [16]. Finnish university lecturers who teach early childhood PE enjoy having autonomy and pedagogical freedom within their syllabuses to organise the teaching content. In the Finnish school system, PE has been a mandated component since the mid-19th century, and teachers responsible for PE must meet the requirement of the master’s degree qualification in PE [39]. PE in ECEC is considered as a key curriculum learning area, and it is implemented by the ECEC teachers, and therefore, PE courses are included in the ECETT programmes. Indeed, ECEC teachers can be seen as “PE teachers in the early years”.

The Finnish Higher Education Evaluation Council (FINHEEC) has outlined that ECETT should be based on scientific knowledge and research data, especially when developing the content of training programmes [40]. Based on the Act regarding Early Childhood Education and Care (i.e., 24§), providers should reflect and evaluate their activities and participate in the evaluation executed by the researchers outside the organisation [41]. Evaluation promotes ECE quality, identifies operational strengths, highlights development needs, and develops activities [18,40]. Promoting teacher-led PE in early childhood, providing training opportunities for professional development, and enacting teachers’ continuous self-monitoring remain warranted activities to support progression in the quality of PE in ECEC [42]. Bai and colleagues [19] have proposed that educator PA-related behaviour and practices can be improved via a professional development intervention. However, according to the review of Wang and Ha [23], 85% of the studies on the professional development of PE teachers have concentrated on in-service teachers, and therefore, there is a need to examine the influence of personal and contextual factors on pre-service teachers’ development. Finally, Howell and Sääkslahti [15] recommend that further accredited education and training is needed to upskill all countries to the same level of knowledge and understanding of PA. This study aimed to investigate pre-service teachers’ perceptions of perceived competence when (1) supporting a child’s PA, (2) teaching PE, and (3) observing and assessing a child’s motor skills and PA. These self-evaluations will be compared with a range of individual, educational, and behavioural characteristics.

## 2. Materials and Methods

### 2.1. Study Participants and Protocol

Two hundred and seventy-four (of a potential 509 invited pre-service teachers) final-year Bachelor degree pre-service teachers from seven universities in Finland (i.e., University of Helsinki, University of Turku, University of Tampere, University of Jyväskylä, University of Eastern Finland, Åbo Akademi, and University of Oulu) completed the self-report questionnaire (response rate of 54%). Responses were provided either online (49%, *n* = 134) using mobile phones, tablets, or computers, or in paper form (51%, *n* = 140), using Finnish (98%, *n* = 269) or English (2%, *n* = 5) versions. Participants took approximately 15 min to complete the questionnaire. Pre-service teachers responded to the questionnaire as part of their ordinary course of study or in their leisure time. Participation in the study was voluntary, and participants could withdraw at any time. No personal information, except age, gender, and nationality, was requested within the questionnaire. Eligibility for involvement in the research required prior completion of compulsory PE studies. A group of 13 respondents were excluded from the final data due to incomplete PE studies, and two respondents did not fully complete the questionnaire.

### 2.2. Ethical Consideration

University directors gave permissions for data collection, and respondents gave their assent by answering the questionnaire. Respondents were informed and allowed to see the privacy statement required by the University of Jyväskylä, where the study was conducted. All material was collected, stored, analysed, and reported so that no participants were identifiable. Researchers used and handled the material’s storage, according to the University of Jyväskylä Ethical Committee guidelines. The data were stored on the University of Jyväskylä’s password-protected server.

### 2.3. Instrument Development and the Variables

A literature search did not identify previous studies related to PE in ECETT or the existence of published early childhood PE questionnaires. When developing the questionnaire for this study, research from school-focussed PETE areas were utilised see, e.g., [43,44,45]. The questions of the current instrument were developed to reflect the specific aims of this research. The questions’ contents were based on representation of the objectives and requirements of ‘The National Core Curriculum for Early Childhood Education and Care 2018’ and ‘Recommendations for Physical Activity in Early Childhood 2016’.

The questionnaire consisted of four parts as follows; (1) respondent’s background information (8 items), (2) respondent’s PA behaviour (7 items; IPAQ-short), (3) respondent’s education and PE studies (8 items), and (4) perceived importance of PE content in ECEC and perceived competence to teach PE (5 items). The questionnaire included 28 questions, of which nine were multiple-choice questions, three were yes/no answers, six were Likert scale, and 10 were open-ended questions. The following detail outlines the variables included in the present study.

Individual characteristics. The participants’ background information including age in years, gender (female, male, transgender, other, do not want to define), and the university where currently studying (University of Helsinki, University of Turku/Rauma Campus, University of Tampere, University of Jyväskylä, University of Eastern Finland, Åbo Akademi, University of Oulu) was requested.

Relevant PE studies. Participants were asked to evaluate, from 1 = completely disagree to 5 = completely agree, their experience of the adequacy of their PE studies in ECETT programmes (11 items; e.g., “I have adequate training in my PE studies to teach PE in ECEC” and “The overall content of my PE studies was adequate”). The subscale for the adequacy of PE studies showed satisfactory reliability (Cronbach’s alpha coefficient (α) = 0.81). The responses format used a five-point (1 = completely disagree, 5 = completely agree) Likert scale to investigate how important/useful/interesting participants thought PE studies are in their ECETT. The subscale of value of PE studies showed a satisfactory reliability (α = 0.74). Furthermore, participants were asked if they have studied PE as a major or minor subject (yes or no options) or have participated in any additional training that has supported them in teaching PE at their work. Responses were obtained using a multiple-choice question. These categories were (1) under 20 h, (2) 20 h or more, or (3) no additional training.

Previous experiences. Participants’ response to the ‘enjoyment of schooltime PE’ item was obtained using a 10-point (1 = not enjoyable at all, 10 = extremely enjoyable) Likert scale. Next, participants were asked, using yes or no options, about their sports instruction or coaching experiences involving groups of 0–8-year-olds (e.g., in sports clubs, associations). Finally, a question involving years of work experience in ECEC settings was asked (none; maximum of two years; 2–5 years; 6–10 years; 11–15 years; 16–20 years and more than 20 years).

The perceived competence of PE. Information regarding participants’ skills and competence in PE were requested as well with a five-point (1 = not competent at all; 5 = highly competent) Likert-scale question: “What are your views of your skills/competence when teaching PE in ECEC? Please select the number that best corresponds to your view”. The question included 16 items that have been highlighted in the National Core Curriculum for Early Childhood Education and Care 2018. Results indicated that the perceived competence of PE had a satisfactory reliability. The Cronbach’s alpha coefficients for the subscales were as follows: *supporting a child’s PA* (sub-items 1, 2, 3) (α = 0.56); *teaching PE* (sub-items 4, 5, 6, 7, 8, 9, 10, 11, 12) (α = 0.76); *observing and assessing a child’s motor skills and PA* (sub-items 13, 14, 15, 16) (α = 0.88).

### 2.4. Statistical Analyses

All analyses were performed using SPSS Version 26. The normality of the data was assessed, and descriptive statistics were completed. Descriptive statistics are expressed as means with standard deviations (SD) or 95% confidence intervals (CI) and counts with percentages. Cronbach’s alpha was formulated as a part of the reliability analyses. The *p*-value for this analysis was <0.05.

The one-way ANOVA analyses were used to compare perceived competence supporting a child’s PA, teaching PE, and observing and assessing a child’s motor skills and PA between the pre-service teachers from different Finnish universities. The last phase of analysis incorporated a set of nine standard linear regressions. The standard linear regressions were performed to assess the ability of individual factors, relevant PE studies, and previous experiences to predict pre-service teachers’ perceived competence when implementing PE. In the analyses, age, gender, university, subscale of adequacy of PE studies in ECETT, subscale of value of PE studies, PE as major or minor subject, additional training that has supported to teach PE, the enjoyment of schooltime PE, sports instruction or coaching experience, and work experience in ECEC were *independent variables*. Whereas, the perceived competence subscale of supporting a child’s PA, subscale of teaching PE, and subscale of observing and assessing a child’s motor skills and PA were *dependent variables*. 

The specific predictor variables used within each of the three regression model sets were as follows:

(1)*Individual characteristics* (3 items): age (1 = maximum of 22 years, 2 = 23–29 years, 3 = 30 years or over), gender (females were coded as 0 and males 1), and university (1 = University of Helsinki, 2 = University of Turku, 3 = University of Tampere, 4 = University of Jyväskylä, 5 = University of Eastern Finland, 6 = Åbo Akademi, and 7 = University of Oulu),(2)*Relevant PE studies* (4 items): the subscale of the adequacy of PE studies (1 = completely disagree, 5 = completely agree), the subscale of value of PE studies (1 = completely disagree, 5 = completely agree), PE as a major or minor subject (no coded as 0 and yes 1), and additional training that has supported teaching PE (no additional training was coded as 0 and all additional training 1), and(3)*Previous experiences* (3 items): the enjoyment of schooltime PE (1 = not enjoyable at all, 10 = extremely enjoyable), sports instruction or coaching experience (no experience was coded as 0 and coaching experience 1), and work experience in ECEC (0 = none; 1 = maximum two years, 2 = 2–5 years, 3 = over six years), to predict the perceived competence in supporting a child’s PA, teaching PE and observing and assessing a child’s motors skills and PA (see Figure 1).

A test–retest questionnaire was executed with the first-, second-, and fifth-year pre-service teachers (*n* = 27) from the University of Jyväskylä. The online questionnaire was completed during participants’ free time for an interval of two weeks. Intra-rater reliability was assessed by comparing the completed questionnaire for each respondent at Time 1 and Time 2, and subsequently reporting both single measure intraclass correlation coefficients (ICCs) using a one-way random absolute model for continuous variables, and Cohen’s kappa with percent agreement for subscales. Overall, the test–retest resulted in a moderate agreement for most of the items. ICCs ranged from fair to good (ICC = 0.55 for teaching PE to 0.70 regarding the adequacy of PE studies).

## 3. Results

Details of the mean values, standard deviations, and percentages associated with the independent variables are presented in Table 1.

Pre-service teachers’ perceptions of their perceived competence to support a child’s PA resulted in the highest mean score (*M* = 3.85, *SD* = 0.53), while the second-highest mean was the skills to teach PE (*M* = 3.63, *SD* = 0.51), and the lowest mean score was for observing and assessing children’s motor skills and PA (*M* = 3.41, *SD* = 0.70). One-way ANOVA analyses were conducted to compare the effect of university location on the pre-service teachers’ perceived competence supporting a child’s PA, teaching PE, and observing and assessing a child’s motor skills and PA. An analysis of variance showed that the effect of location of university on perceived competence supporting a child’s PA (*F* (1.388, 71.210) = 0.861, *p* = 0.524), teaching PE (*F* (3.152, 67.729) = 2.032, *p* = 0.062), and observing and assessing a child’s motor skills and PA (*F* (6.031, 125.413) = 2.132, *p* = 0.050) was statistically non-significant (Table 2).

The results for the nine linear regressions of the three linear regression models are reported in Table 3, Table 4 and Table 5. The R squared results ranged from 0.014 to 0.186.

The independent variables that were the strongest predictors of dependent variables within the regression analyses were the subscale of adequacy of PE studies and sports instruction and coaching in predicting supporting a child’s PA (*p* < 0.001); the subscale of adequacy of PE studies and the enjoyment of schooltime PE in predicting teaching PE (*p* < 0.001); and the subscale of adequacy of PE studies in predicting observing and assessing a child’s motor skills and PA (*p* < 0.001).

The independent variables that demonstrated the lowest values for the prediction of dependent variables were PE as a major or minor subject in supporting a child’s PA (*p* = 0.139); the subscale of value of PE studies in teaching PE (*p* = 0.852); and age in observing and assessing a child’s motor skills and PA (*p* = 0.699).

## 4. Discussion

The purpose of this research was to investigate pre-service teachers’ perceptions of the perceived competence to support a child’s PA, teach PE, and observe and assess a child’s motor skills and PA. Overall, pre-service teachers’ perceptions of perceived competence in PE were indicative of a positive level of competence. In addition, perceived competence to support a child’s PA was rated the highest, teaching skills in PE was rated the second highest, and the lowest scores were for observing and assessing a child’s motor skills and PA scale.

Relevant PE studies. Limited information is available concerning teachers’ competence and confidence in PE, including physical literacy knowledge and its application to practice [42]. The present research results support the significance of education in pre-service teachers’ perceptions. The findings highlighted that relevant PE studies, such as adequacy of PE studies in ECETT, predict all pre-service teachers’ perceptions of perceived competencies. Understandably, a person perceives their skills higher the more positively they perceive the adequacy of their studies. Indeed, this information highlights the importance of PE in ECETT programmes, and the time devoted to this syllabus area should be maintained or increased.

Lu and Montague [28] outlined concerns regarding teachers’ lack of adequate training and knowledge to develop and lead structured PA sessions. Furthermore, teachers seem to have limited learning opportunities and practice of PE in actual ECEC settings [42]. In the present study, a person’s additional training in PE predicted higher perceived competence to support a child’s PA and teach PE. Additionally, if a person has had PE as a major or minor subject, he or she was more competent in observing and assessing a child’s motor skills and PA. At the same time, the higher scores in the value of PE predicted an increase in reported competency levels to support a child’s PA. In line with the present study, Bruijns et al. [46] have suggested that providing increased ECEC training opportunities may help promote early childhood teachers’ competence and perceived capability to design and implement PA activities among children. Importantly, it may even lead to more intentional monitoring and programming of PA in early years’ curricula [46]. Professional development programmes need to ensure that all teachers become capable of providing adequate quality PA opportunities for young children [42]. Notably, the need for additional training has been highlighted in places where the higher education curricula for pre-service teachers have limited courses with a particular emphasis on PA, physical literacy, or movement skill development [42]. Consequently, it is recommended that training opportunities linked with continuous follow-up support for teachers should be made available to strengthen pre-service and in-service teachers in integrating teaching strategies that promote children’s PA in ECEC settings [42].

Previous experiences. The present research results strongly show that pre-service teachers’ previous experiences such as sports instruction or coaching, work in ECEC, and memories of the enjoyment of schooltime PE predicted the current perceptions of competence to support, teach, and observe a child’s PA. Specifically, the enjoyment of schooltime PE predicted higher competence supporting, teaching, and observing and assessing a child’s PA. Sports instruction or coaching experiences predicted higher competence to support a child’s PA and teach PE. Incorporating information of pre-service teachers’ experiences in PE and recreational level coaching can be used to support the development of their own PE professional skills. Finally, work experience in ECEC was related to higher competence supporting a child’s PA and observing and assessing a child’s motor skills and PA. Observation is an essential tool of the early childhood educator for evaluating children’s development and skills. However, it can be seen as a demanding method that requires substantial professional experience because observation skills evolve with the engagement within the ECEC settings [47].

Previous studies have shown that tertiary-level qualified teachers have been associated with reports of higher levels of PA among the children attending ECEC settings in which they work [20,21,22]. Consistent with earlier findings, the current results showed the potential of ECETT programmes in promoting more positive perceptions of pre-service teachers’ perceived competence in PE. Indeed, access to greater opportunities to engage in training and education in PE can positively influence a person’s attitudes and perceptions of teaching PE [23]. For instance, informing early childhood teachers on the benefits of PA can change their perceptions by obtaining more positive attitudes towards children’s PA [21].

Furthermore, Riley et al. [48] has highlighted that programmes offering professional development have positively influenced teachers’ perceptions towards PA. Moreover, teachers’ positive perceptions towards PA are requisite for promoting children’s PA participation [21] and may bring children closer to meeting the daily PA recommendations. Programmes designed to increase children’s PA are likely to be attractive to teachers if they are designed in partnership with education authorities [48]. According to Trost et al. [22], ECEC settings with increased levels of children’s PA levels tend to employ teachers with higher levels of education. Therefore, ECETT programmes play a central role in teachers’ fundamental professional development, and the training of early childhood teachers in all aspects of PA is important. Notably, in Finnish ECETT, pre-service teachers are provided with training focussing on theoretical knowledge, such as the benefits of versatile PA and practical possibilities of PA implementations in ECEC settings.

Individual characteristics. Although previous studies have stated differences in teaching habits between male and female teachers see e.g., [32], only minor gender differences were observed in the present study. Male pre-service teachers’ perceived higher competence teaching PE than female pre-service teachers. In the present study, there were only 22 male (8%) participants. While in Finland in 2015, from 16,201 ECEC teachers, 97% were women and only 3% were male teachers [49]. The present sample size closely reflects the gender patterns currently observed in the workforce. Furthermore, pre-service teachers’ age or the location of the university did not predict pre-service teachers’ perceived competence in PE. Nevertheless, the finding did not reflect the observed differences in the execution of PE in different Finnish universities (e.g., PE syllabuses, amount and content of PE studies, and different implementation methods). One explanation for the similarity is that the National Core Curriculum for Early Childhood Education and Care [18] is a mandated document that guides the content of ECEC settings and ECETT programmes. The consistency in pre-service teachers’ perceptions of PE in ECETT is reinforced through the Finnish universities ECETT networks sharing professional knowledge, common goals, and course content.

The review of Wang and Ha [23] revealed a clear knowledge gap in the research on PE teacher development, and more studies are needed to address the professional development of pre-service teachers. In addition, previous studies in PETE have indicated the positive relationship between professional development programmes and teachers’ knowledge, beliefs, and teaching behaviour on students’ learning performance. Tortella et al. [16] highlight that ECETT programmes should include outdoor activities as one didactic method to support academic learning and PA to promote a child’s holistic development. Furthermore, over a three-year teacher training period, it is possible to strengthen the skills of teachers, regardless of the participants’ starting situation. As previously described, early childhood teachers’ high-quality PE education supports the opportunity for children to regularly access PE and PA. Moreover, PE makes it possible that children have a positive movement experience and increase their overall PA and well-being. Therefore, the current study findings play an important role in developing ECETT programmes and supporting the future ECEC teachers’ professional skills in implementing PE in the early years.

Overall, the strength of this study is its focus on PE in ECETT. Notably, the sample consisted of participants from all seven universities that arrange ECETT in Finland. However, some limitations need to be acknowledged. While no previous research or questionnaires in ECETT’s PE were available, it was necessary to develop a specific questionnaire for the current research. In the future, the questionnaire should be psychometrically evaluated as part of future measure development. The current and future versions should be easily modifiable to a range of languages and cultures and therefore available for use in comparative studies.

Although no variations were found between pre-service teachers’ perceptions in Finnish universities, curriculums and ECETT programmes vary from country to country; see, e.g., [15,16]. For instance, in Nordic countries, children’s exposure to outdoor environments is included in teacher education programmes and national curriculums [16]. Consideration of this approach may contribute to positive changes in existing international curriculums that strengthen opportunities for outdoor movement education and active outdoor play. Additionally, if the PE teacher development programme aims to be effective, professional development should be considered through multiple lenses and aligned with district policy and curriculum requirements [23]. More extensive practical trials are required to confirm the impact of professional development programmes on teachers’ perceived PA behaviour [19]. Indeed, in the future, it is recommended that support is accessible to all countries in furthering knowledge of PE in ECETT programmes [15]. Moreover, research should address the standardising of methods for data collection and reporting data to achieve more reliable results and improved comparability of findings between studies and countries.

## 5. Conclusions

The study findings demonstrated that education can positively influence perceptions and attitudes to increase a pre-service teacher’s perceived competence when implementing PE in the early years. Furthermore, the current results highlighted how relevant PE studies and previous experiences can predict a person’s perceived competencies and skills in PE. Policies, curriculum, and ECETT programmes vary from country to country; therefore, international comparisons and curriculum reviews across the subject domain are warranted.

## Figures and Tables

**Figure 1 ijerph-18-06454-f001:**
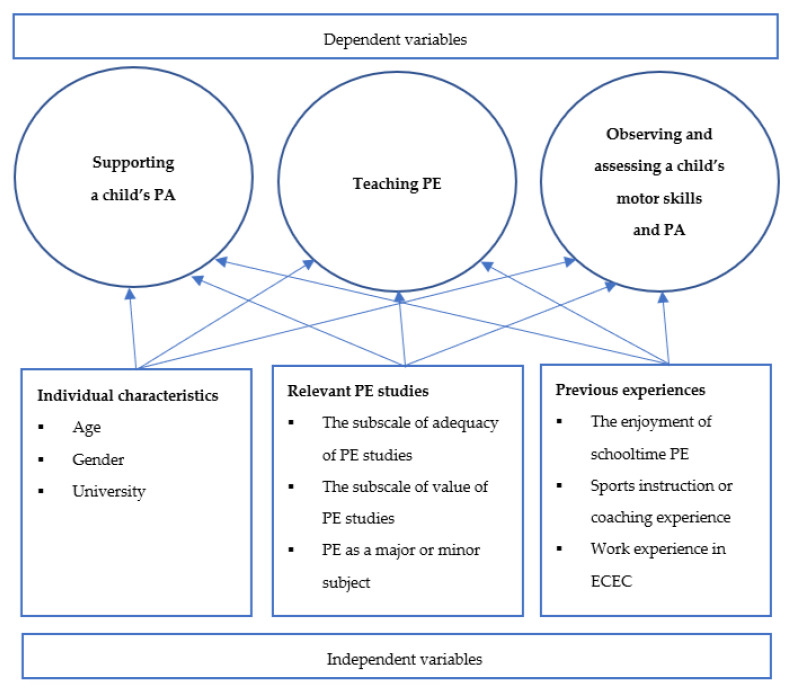
The conceptual model of the variables.

**Table 1 ijerph-18-06454-t001:** Demographic and variable characteristics.

		Individual Characteristics							Relevant PE Studies			Previous Experiences						
			Gender		Age				Adequacy of PE Studies (1–5)	The Value of PE Studies (1–5)	PE as a Major or Minor Subject (yes)	Additional PE Training (yes)	School Time PE (1–10)	Sports Coaching (yes)	Work Experience in ECEC in Years			
		All	Female	Male	*n* = 274	≤22	23–29	≥30	*n* = 265	*n* = 269	*n* = 274	*n* = 274	*n* = 265	*n* = 272	*n* = 274			
University		*n* (%)	*n* (%)	*n* (%)	*M* (*SD*)	*n* (%)	*n* (%)	*n* (%)	*M* (*SD*)	*M* (*SD*)	*n* (%)	*n* (%)	*M* (*SD*)	*n* (%)	None (%)	≤2 (%)	2–5 (%)	≥6 (%)
University of Helsinki		39 (14)	36 (92)	3 (8)	30.87 (8.13)	6 (15)	12 (31)	21 (54)	3.20 (0.49)	4.76 (0.34)	0 (0)	7 (18)	6.61 (2.52)	11 (28)	15	49	26	10
University of Turku		35 (13)	35 (100)	0 (0)	25.31 (5.13)	11 (31)	18 (51)	6 (18)	3.43 (0.44)	4.61 (0.59)	4 (11)	9 (26)	7.91 (1.60)	11 (32)	14	63	23	0
University of Tampere		45 (16)	43 (96)	2 (4)	25.87 (6.55)	21 (47)	14 (31)	10 (22)	3.15 (0.49)	4.63 (0.51)	4 (9)	7 (16)	7.20 (2.06)	11 (24)	42	42	16	0
University of Jyväskylä		65 (24)	60 (92)	5 (8)	25.05 (5.41)	30 (46)	27 (42)	8 (12)	3.50 (0.40)	4.82 (0.35)	4 (6)	8 (12)	7.56 (2.20)	15 (23)	28	60	8	5
University of Eastern Finland		32 (12)	26 (81)	6 (19)	25.78 (6.63)	11 (34)	15 (47)	6 (19)	3.29 (0.46)	4.67 (0.57)	0 (0)	4 (13)	7.90 (1.61)	14 (44)	34	50	9	6
Åbo Akademi		12 (4)	11 (92)	1 (8)	22.25 (1.49)	8 (67)	4 (33)	0 (0)	3.47 (0.51)	4.53 (0.61)	1 (8)	2 (17)	6.45 (2.54)	5 (42)	8	58	25	8
University of Oulu		46 (17)	41 (89)	5 (11)	24.48 (4.41)	19 (41)	22 (48)	5 (11)	2.88 (0.46)	4.59 (0.50)	5 (11)	1 (2)	6.80 (2.70)	8 (18)	26	70	2	2
Total		274	252 (92)	22 (8)	25.91 (6.26)	106 (39)	112 (41)	56 (20)	3.25 (0.50)	4.68 (0.48)	18 (7)	38 (14)	7.27 (2.20)	75 (28)	26	56	14	4

**Table 2 ijerph-18-06454-t002:** A comparison of pre-service teachers’ perceived PE competence in different Finnish universities.

Perceived PE Competencies	Total	A	B	C	D	E	F	G	ANOVA *p*-Value	Scheffe	Eta^2^
	*n* = 271	*n* = 39	*n* = 34	*n* = 44	*n* = 65	*n* = 31	*n* = 12	*n* = 46			
	*M* (*SD*)	*M* (*SD*)	*M* (*SD*)	*M* (*SD*)	*M* (*SD*)	*M* (*SD*)	*M* (*SD*)	*M* (*SD*)			
Supporting a child’s PA	3.85 (0.53)	3.92 (0.50)	3.97 (0.53)	3.87 (0.51)	3.77 (0.46)	3.85 (0.54)	3.69 (0.66)	3.83 (0.55)	0.524	0.633	0.019
Teaching PE	3.63 (0.51)	3.52 (0.43)	3.83 (0.47)	3.53 (0.59)	3.69 (0.52)	3.73 (0.40)	3.56 (0.47)	3.55 (0.56)	0.062	0.465	0.044
Observing and assessing a child’s motor skills and PA	3.41 (0.70)	3.33 (0.62)	3.69 (0.56)	3.21 (0.87)	3.45 (0.72)	3.44 (0.49)	3.65 (0.69)	3.33 (0.68)	0.050	0.265	0.046

A = University of Helsinki; B = University of Turku; C = University of Tampere; D = University of Jyväskylä; E = University of eastern Finland; F = Åbo Akademi; G = University of Oulu.

**Table 3 ijerph-18-06454-t003:** Linear regression results for supporting a child’s PA.

Regression Model	Independent Variables	Coefficient (Std. Error)	*p*-Value	Constant	R^2^	F-Ratio
**Individual characteristics**						
	Age	0.022 (0.043)	0.613	3.769	0.014	1.232
	Gender	0.208 (0.116)	0.075			
	University	0.007 (0.019)	0.699			
**Relevant PE studies**						
	The subscale of adequacy of PE studies	0.223 (0.062)	<0.001	2.425	0.123	8.883
	The subscale of value of PE studies	0.139 (0.068)	0.042			
	PE as a major or minor subject	0.226 (0.129)	0.081			
	Additional training	0.197 (0.091)	0.032			
**Previous experiences**						
	The enjoyment of schooltime PE	0.046 (0.014)	0.001	3.334	0.149	15.008
	Sports instruction or coaching experience	0.306 (0.068)	<0.001			
	Work experience in ECEC	0.086 (0.039)	0.027			

**Table 4 ijerph-18-06454-t004:** Linear regression results for teaching PE.

Regression Model	Independent Variables	Coefficient (Std. Error)	*p*-Value	Constant	R^2^	F-Ratio
**Individual characteristics**						
	Age	−0.037 (0.042)	0.379	3.634	0.019	1.768
	Gender	0.241 (0.114)	0.036			
	University	0.012 (0.019)	0.504			
**Relevant PE studies**						
	The subscale of adequacy of PE studies	0.381 (0.059)	<0.001	2.287	0.186	14.496
	The subscale of value of PE studies	0.012 (0.064)	0.852			
	PE as a major or minor subject	0.119 (0.123)	0.334			
	Additional training	0.233 (0.087)	0.008			
**Previous experiences**						
	The enjoyment of schooltime PE	0.057 (0.014)	<0.001	3.091	0.110	10.651
	Sports instruction or coaching experience	0.178 (0.070)	0.011			
	Work experience in ECEC	0.066 (0.040)	0.099			

**Table 5 ijerph-18-06454-t005:** Linear regression results for observing and assessing a child’s motor skills and PA.

Regression Model	Independent Variables	Coefficient (Std. Error)	*p*-Value	Constant	R^2^	F-Ratio
**Individual characteristics**						
	Age	0.022 (0.058)	0.699	3.307	0.021	1.880
	Gender	0.301 (0.156)	0.054			
	University	0.038 (0.025)	0.137			
**Relevant PE studies**						
	The subscale of adequacy of PE studies	0.409 (0.085)	<0.001	2.300	0.119	8.590
	The subscale of value of PE studies	−0.060 (0.093)	0.521			
	PE as a major or minor subject	0.484 (0.177)	0.007			
	Additional training	0.117 (0.125)	0.351			
**Previous experiences**						
	The enjoyment of schooltime PE	0.063 (0.019)	0.001	2.730	0.094	8.914
	Sports instruction or coaching experience	0.169 (0.096)	0.079			
	Work experience in ECEC	0.176 (0.055)	0.001			

## Data Availability

The data presented in this study are available on request from corresponding author.

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
