# Peer review of "Finnish Pre-Service Teachers’ Perceptions of Perceived Competence in Early Childhood Physical Education"

_ijerph, 2021, doi:10.3390/ijerph18126454_

Round 1

Reviewer 1 Report

Dear author(s), thanks for submitting your manuscript. I think that analyzing perceptions of pre-service teachers perceived competence in early childhood PE will add innovation and relevance to the field of physical education/physical activity in early years literature, and hope this brings a positive impact in the lives of young children.

I made some suggestions regarding readability of your manuscript.

  • Line 64 –remove ‘with educators whilst attending ECEC’ and add ‘with them’
  • Line 67 – Add the word ‘physical’ before ‘activity’
  • Line 70 – Remove “in addition’ and start a new paragraph with ‘Previous studies…’. The paragraph (lines 59-89)is too long.
  • Line 84 – remove ‘they’ and add he/she to be consistent within the sentence pronouns.
  • Line 92 – remove ‘and for instance,’ and add ‘consequently teachers…’
    • Remove ‘the’
  • Line 97 – Instead of ‘In Finland PE’ use ‘PE in Finland’. Please be specific if this PE is in Early Childhood.
  • Line 99 – Remove ‘it has been shown’ and add ‘a study has shown and include the reference.
  • Line 105 – I suggest finding a substitute word for ‘carers.’ For example a ‘child care provider.’
  • Line 105 – Think about adopting one type of terminology across the manuscript. For example, teachers, educators or pedagogues.
  • Line 115 – remove ‘for instance’
  • Line 123 – replace teachers by professors or instructors.
  • Line 123 – rearrange the sentence. For example: ‘Finnish university professors who teach early childhood PE enjoy having autonomy and pedagogical freedom within their syllabuses to organize the teaching content.
  • Line 126 – replace ‘met’ by ‘meet’
  • Line 128 – remove ‘programs’ and add ‘courses’
  • Line 132 – remove ‘in Finland’ -It’s not necessary
  • Line 145 – remove the word ‘strongly’
  • Line 157 – remove ‘participated in the completion’ and add ‘responded’ the self-report …
  • Line 160 – replace ‘Completing the questionnaire took approximately 15 minutes’ by Participants took approximately 15 minutes to complete the questionnaire.
  • Line 161 – remove ‘completed’ add ‘responded to’
  • Lines 311-317 – font size is different from the previous text
  • Line 357 – Predicted
  • Lines 358-360 – These sentences need revision. The word ‘whereas’ in the beginning of the sentence made it confusing.
  • I suggest organizing your discussion by somehow categorizing your findings (e.g., perceived competence). This will help the reader to locate specific information.

Reviewer 2 Report

A major strengh of the paper is its succint Introduction. However, please highlight why studying pre-service teachers' role is important. Just because previous studies focused primarily on in-service teachers by itself is not sufficient as a justification for the inclusion of pre-service teachers. There must be some practical/theoretical importance that goes beyond the gap in the literature. If there is any, please be sure to explain early on in the Introduction. Also, it would be helpful to include a Figure (e.g., graph, conceptual model) to illustrate the relationships among variables that are under investigation.
